# Peripheral blood CD4+CCR6+ compartment differentiates HIV-1 infected or seropositive elite controllers from long-term successfully treated individuals

Sara Svensson Akusjärvi [1✉], Shuba Krishnan[1], Bianca B. Jütte[2], Anoop T. Ambikan[1], Soham Gupta [1], Jimmy Esneider Rodriguez [3], Ákos Végvári [3], Maike Sperk[1], Piotr Nowak[4], Jan Vesterbacka[4], J. Peter Svensson [2], Anders Sönnerborg[1,4] & Ujjwal Neogi [1,5,6✉]

HIV-1 infection induces a chronic inflammatory environment not restored by suppressive antiretroviral therapy (ART). As of today, the effect of viral suppression and immune reconstitution in people living with HIV-1 (PLWH) has been well described but not completely understood. Herein, we show how PLWH who naturally control the virus (PLWH$_{EC}$) have a reduced proportion of CD4+CCR6+ and CD8+CCR6+ cells compared to PLWH on suppressive ART (PLWH$_{ART}$) and HIV-1 negative controls (HC). Expression of CCR2 was reduced on both CD4+, CD8+ and classical monocytes in PLWH$_{EC}$ compared to PLWH$_{ART}$ and HC. Longer suppressive therapy, measured in the same patients, decreased number of cells expressing CCR2 on all monocytic cell populations while expression on CD8+ T cells increased. Furthermore, the CD4+CCR6+/CCR6− cells exhibited a unique proteomic profile with a modulated energy metabolism in PLWH$_{EC}$ compared to PLWH$_{ART}$ independent of CCR6 status. The CD4+CCR6+ cells also showed an enrichment in proteins involved in apoptosis and p53 signalling in PLWH$_{EC}$ compared to PLWH$_{ART}$, indicative of increased sensitivity towards cell death mechanisms. Collectively, this data shows how PLWH$_{EC}$ have a unique chemokine receptor profile that may aid in facilitating natural control of HIV-1 infection.

[1] Division of Clinical Microbiology, Department of Laboratory Medicine, Karolinska Institutet, ANA Futura, Campus Flemingsberg, 141 52 Stockholm, Sweden. [2] Department of Biosciences and Nutrition, Karolinska Institutet, Neo, Campus Flemingsberg, 141 83 Stockholm, Sweden. [3] Division of Chemistry I, Department of Medical Biochemistry and Biophysics, Karolinska Institutet, Campus Solna, 171 65 Stockholm, Sweden. [4] Division of Infectious Disease, Department of Medicine Huddinge, Karolinska Institutet, I73, Karolinska University Hospital, 141 86 Stockholm, Sweden. [5] Christopher S. Bond Life Sciences Centre, University of Missouri, Columbia, MO 65211, USA. [6] Manipal Institute of Virology (MIV), Manipal Academy of Higher Education, Manipal, Karnataka, India. ✉email: sara.svensson.akusjarvi@ki.se; ujjwal.neogi@ki.se

Efficient antiretroviral therapy (ART) suppresses human immunodeficiency virus type 1 (HIV-1) replication to below the detection level of sensitive HIV RNA assays. Still, HIV-1 persists for decades despite efficient ART in presumably latently infected cells, making up the so-called latent viral reservoir. This reservoir resides in subtypes of memory T cells and cells from monocytic lineages, but also in some functional and naïve T cell subsets[1]. Eradication of latently infected cells, i.e. a sterilizing cure, has proven to be a major hurdle due to several factors such as heterogeneity in mechanisms governing latency and integration sites into the host genome[2]. Over the course of suppressive ART, persisting HIV-1 sustained by low levels of viral replication, homoeostatic proliferation, and cell-to-cell transmission, is a driver of chronic immune activation in the host[3–5].

In HIV-1 infection, chemokine receptors are essential co-receptors needed for HIV-1 entry into the cell where the CC chemokine receptor 5 (CCR5) and CXC chemokine receptor 4 (CXCR4) are the main co-receptors used for HIV-1 entry, but alternative receptors such as CCR3 may play a role for macrophage-tropic viral strains[6–8]. T cells expressing the chemokine receptor CCR6 are also overrepresented amongst cell types infected by HIV-1[9]. Furthermore, $CD4^+CCR6^+$ and $CD4^+CCR2^+CCR5^+$ cells have been proposed to contribute to the HIV-1 reservoir[10,11].

Alterations in chemokine receptor expression and chemokine levels modulate the activity of the immune response and inflammatory levels during HIV-1 infection[12]. In people living with HIV-1 (PLWH) on suppressive ART, these aberrations may contribute to (i) higher risk of age-associated co-morbidities together with elevated immune activation, (ii) microbiome dysbiosis leading to microbial translocation and (iii) dysregulated immune cell activation and function[13–16]. Consequently, HIV-1 pathogenesis and immune dysfunction promote chronic immune activation. This results in a vicious cycle as immune activation likely drives HIV-1 disease progression and inflammaging, defined as an age-related increase of inflammatory markers, that could possibly lead to earlier onset of age-related diseases. In our earlier study on a small group of 10 PLWH on ART for two decades, we reported normalisation of most of the plasma proinflammatory cytokines and chemokines to the level of HIV-negative controls[17]. Therefore, although there are several studies on immune reconstitution and viral suppression during short-term ART, the long-term effect of ART on the immune system is not known.

Eradication of latently infected cells, i.e., a sterilising cure, has proven to be a major hurdle. An alternative approach is a functional cure aimed at suppressing viral replication without ART. A model for functional cure studies is elite controllers (herein $PLWH_{EC}$). $PLWH_{EC}$ constitute a small fraction (<0.5%) of HIV-$1^+$ individuals able to naturally control viral replication in the absence of ART[18,19]. However, there is a population-based heterogeneity of how these individuals suppress viral replication which is likely caused by several factors such as viral and host genetic factors, variability of integration site in the human genome, and variation in the host response such as immunological factors[20–23]. In terms of immune cell activation, studies have shown how $PLWH_{EC}$ maintain lower levels of inflammatory markers compared to PLWH who progress in their disease[24]. However, as no consensus exists on how the $PLWH_{EC}$ phenotype is maintained, further studies comparing variability of immune function between $PLWH_{EC}$ and PLWH on suppressive ART (herein $PLWH_{ART}$) are required to understand the mechanisms of natural control of HIV-1.

In our recent study[25], we hypothesised that an altered CCR6/CCL20 chemokine axis and CCR2-CCL7-CCL2 signalling may play a protective role in the $PLWH_{EC}$ phenotype. We identified a

downregulation of CCR2 and CCR6 receptors on $CD4^+$ and $CD8^+$ lymphocytes and higher plasma abundance of CCL4, CCL7 and CCL20 in $PLWH_{EC}$ compared to the HIV-1-negative controls. This can provide natural resistance to HIV-1 infection although the expression profile during long-term suppressive ART is not known. In the present study, we extended our analysis to evaluate the expression profile dynamics of key chemokine receptors CCR2, CCR3, CCR5 and CCR6 in $PLWH_{ART}$. We also compared the proportion of integrated HIV-1 between $PLWH_{EC}$ and $PLWH_{ART}$. Furthermore, cell populations of interest were isolated for quantitative proteomics to evaluate specific characteristics regulating these cell populations and their potential role in HIV-1 persistence. Our study provides important understanding of the chemokine receptor dynamics and its role in HIV-1 persistence that differentiate $PLWH_{EC}$ from $PLWH_{ART}$.

## Results

**$PLWH_{EC}$ have reduced populations of $CD4^+CCR6^+$ and $CD4^+CCR2^+$ T lymphocytes compared to successfully long-term treated patients.** Herein, we wanted to evaluate if there are any differences in the expression profile of some key chemokine receptors between PLWH with natural control of the virus and those on ART. Therefore, we investigated the expression levels of CCR2, CCR3, CCR5 and CCR6 in peripheral blood mononuclear cells (PBMCs) from $PLWH_{EC}$ ($n = 14$), $PLWH_{ART}$ ($n = 54$) and HIV-negative individuals (HC, $n = 18$) by flow cytometry (Supplementary Fig. 1). The clinical and demographic data are presented in Table 1 and Supplementary Table 1. All the $PLWH_{ART}$ were successfully treated at the time of sample collection and plasma viral load was below the detection level (<40 copies/mL). Initial discrimination between $CD4^+$ T cells ($CD3^+CD4^+$) and $CD8^+$ T cells ($CD3^+CD8^+$) showed that both $PLWH_{EC}$ and $PLWH_{ART}$ exhibited a reduced proportion of $CD4^+$ T cells and elevated proportion of $CD8^+$ T cells compared to HC (Fig. 1a). Single receptor expression in $PLWH_{EC}$ showed a distinct profile compared to both $PLWH_{ART}$ and HC (Fig. 1b). Expression of CCR2 and CCR6 was reduced on both $CD4^+$ and $CD8^+$ T cells together with a decreased proportion of $CD8^+CCR2^+$, $CD4^+CCR6^+$ and $CD8^+CCR6^+$ in $PLWH_{EC}$ compared to both $PLWH_{ART}$ and HC (Fig. 1c, d). The expression levels of CCR2 were also reduced on $CD8^+$ T cells in $PLWH_{ART}$ compared to HC, as well as the proportion of $CD8^+CCR6^+$ cells (Fig. 1c, d). Even as the total frequency of $CD4^+CCR3^+$ was low in all three groups, the expression of CCR3 was high in both $PLWH_{EC}$ and $PLWH_{ART}$ compared to HC (Supplementary Fig. 2a). Furthermore, expression levels of CCR5 were lower on $CD8^+$ T cells in $PLWH_{EC}$ compared to $PLWH_{ART}$ (Supplementary Fig. 2b), while no difference was detected on $CD4^+$ T cells. Collectively, these data show how $PLWH_{EC}$ have a unique expression profile of CCR2 and CCR6 while the expression signature of CCR3 is more similar to $PLWH_{ART}$ on lymphocytic cells (Fig. 1e). Continuing, co-expression analysis on $CD4^+$ cells showed that $CCR2^+CCR5^+CCR6^+$, $CCR2^+CCR6^+$ and $CCR5^+CCR6^+$ were reduced in $PLWH_{EC}$ compared to both $PLWH_{ART}$ and HC, while no differences were detected between $PLWH_{ART}$ and HC (Fig. 1f). Furthermore, single receptor expression of CCR2 was reduced in $PLWH_{EC}$ compared to $PLWH_{ART}$ and CCR6 expression was reduced in $PLWH_{EC}$ compared to both $PLWH_{ART}$ and HC (Fig. 1f). Within the $CD8^+$ T cell population, co-receptor expression of $CCR2^+CCR5^+CCR6^+$ was reduced in both $PLWH_{ART}$ and $PLWH_{EC}$ compared to HC while $CCR2^+CCR5^+$ was reduced in $PLWH_{EC}$ compared to HC and $PLWH_{ART}$ (Fig. 1g). Single receptor expression analysis also showed that CCR5 was increased in $PLWH_{EC}$ compared to HC and CCR6 expression decreased in $PLWH_{EC}$ compared to both HC and

**Table 1 Patient characteristics.**

| Parameter | PLWH$_{EC}$ | PLWH$_{ART}$ | HC | P-values |
|---|---|---|---|---|
| N | 14 | 54 | 18 | ND |
| Gender: Female, N (%) | 7 (50) | 19 (35) | 9 (50) | ND |
| Ethnicity, N (%) | | | | |
| Black | 9 (64.2) | 21 (38.8) | 5 (27.7) | ND |
| Asian | – | 2 (3.7) | – | |
| Caucasian | 4 (28.6) | 31 (57.4) | 13 (72.2) | |
| Hispanic | 1 (7.1) | – | – | |
| *At sampling* | | | | |
| Age in years, mean (SD) | 46 (10.5) | 50 (9.5) | 47 (9) | 0.2704[a] |
| CD4$^+$ T cell count (cells/µL); median (IQR) | 680 (570-1060) | 660 (485-795) | NA | 0.5012[b] |
| CD8$^+$ T cell count (cells/µL); median (IQR) | 900 (540-1330) | 600 (415-950) | NA | 0.0974[b] |
| CD4:CD8 ratio; median (IQR) | 1 (0.44-1.18) | 1 (0.7-1.45) | NA | 0.3295[b] |
| Duration of treatment in years; median (IQR) | NA | 8 (6.75-19) | NA | ND |
| Duration of suppressive treatment in years; median (IQR) | NA | 8 (7-15) | NA | ND |
| Years HIV positive; median (IQR) | 7 (4.5-11.7) | 12.5 (9-20) | NA | 0.0012[b] |
| HIV RNA load; | <40 copies/mL | <40 copies/mL | NA | ND |
| Copies/mL (IQR) | | | | |
| Treatment regimen, N (%) | | | | |
| 3rd drug, | | | | |
| Boosted PI | | 2 (3.7) | | |
| INSTI | | 30 (55.5) | | |
| NNRTI | NA | 21 (38.8) | NA | NA |
| 1st drug, | | | | |
| ABC | | 34 (62.9) | | |
| TAF/TDF | | 16 (29.6) | | |
| Other | | 4 (7.4) | | |
| *Initiation of treatment* | | | | |
| CD4$^+$ T cell count at treatment initiation (cells/µL); median (IQR) | NA | 351 (167.5-522.5) | NA | ND |
| Viral Load at treatment initiation; Log$_{10}$ copies/mL (IQR) | NA | 4.79 (4.06-5.44) | NA | ND |

*ABC abacavir, IQR interquartile range, N number, NA not applicable, ND not done, PI protease inhibitor, INSTI integrase strand transfer inhibitor, NNRTI non-nucleoside reverse transcriptase inhibitor, SD standard deviation, TAF tenofovir alafenamide, TDF tenofovir disoproxil.*
[a]One-way ANOVA.
[b]Mann–Whitney test.

PLWH$_{ART}$ (Fig. 1g). As the major differences were detected on CCR2 and CCR6 expression, we measured the levels of their ligands, namely monocyte chemoattractant protein 1, MCP1 (CCL2) and macrophage inflammatory protein 3, MIP-3 (CCL20), respectively, in plasma from PLWH$_{ART}$ ($n = 49$), PLWH$_{EC}$ ($n = 12$) and HC ($n = 16$). Plasma levels of CCL20 were increased in both PLWH$_{EC}$ and PLWH$_{ART}$ compared to HC, while no significant difference was detected for CCL2 (Supplementary Fig. 2c, d). In summary, the receptor expression profile on lymphocytes in PLWH$_{ART}$ resembles HC, while PLWH$_{EC}$ exhibits a distinct expression profile with decreased receptor expression of both CCR2 and CCR6.

**CCR2 expression is reduced in PLWH$_{EC}$ compared to PLWH$_{ART}$ on monocytic cell populations.** In HIV-1 infection there is a chronic activation of monocytes contributing to organ-specific inflammatory events that are poorly understood in virologically suppressed patients. Our group and others have previously shown persistent low-level inflammation in PLWH$_{EC}$[24–26]. Furthermore, chemokine receptors like CCR2 and CCR5 play a major role in inflammatory trafficking of monocytes[27]. Therefore, we determined the CCR2, CCR3, CCR5 and CCR6 receptor expression in monocytic cell populations; classical (CM, CD14$^+$CD16$^-$), intermediate (IM, CD14$^+$CD16$^+$) and non-classical (NCM, CD14$^-$CD16$^+$). Herein, PLWH$_{EC}$ exhibited reduced numbers of CM compared to HC and elevated levels of IM compared to both HC and PLWH$_{ART}$ (Fig. 2a, b and Supplementary Fig. 3a). Generally, the receptor expression of

CCR3 and CCR6 was low on all three monocytic cell subsets (Fig. 2c). The major difference was a reduced expression of CCR2 on CM and IM in PLWH$_{EC}$ compared to HC and on CM compared to PLWH$_{ART}$ (Fig. 2d). In PLWH$_{ART}$, both the frequency of IM-CCR2$^+$ cells and expression of CCR2 on IM cells were reduced compared to HC (Fig. 2d). Although the proportion of CM-CCR6$^+$ cells was decreased in PLWH$_{EC}$ compared to PLWH$_{ART}$, CCR6 expression was increased in PLWH$_{EC}$ compared to HC and PLWH$_{ART}$, while PLWH$_{ART}$ showed reduced expression compared to HC (Supplementary Fig. 3b). Furthermore, NCM-CCR6$^+$ cells were lower in PLWH$_{EC}$ compared to PLWH$_{ART}$ and HC, and the frequency of CM-CCR3$^+$ cells was decreased in PLWH$_{EC}$ compared to PLWH$_{ART}$ (Supplementary Fig. 3b, c). The receptor expression of CCR5 was reduced on IM in PLWH$_{ART}$ compared to HC and on NCM in PLWH$_{EC}$ compared to both PLWH$_{ART}$ and HC (Supplementary Fig. 3d). This data indicates that the main difference in the monocytic cell subsets is a reduced expression of CCR2 in PLWH$_{EC}$ compared to the other groups (Fig. 2e). The IM were the only monocytic cell population which exhibited differences in co-receptor expression. For these cells, the largest variation was a reduction of CCR2$^+$CCR5$^+$ expressing cells in both PLWH$_{EC}$ and PLWH$_{ART}$ compared to HC. An increase in cells lacking all four receptors was found in both PLWH$_{EC}$ and PLWH$_{ART}$ compared to HC (Fig. 2f). Furthermore, PLWH$_{ART}$ exhibited reduced numbers of cells expressing only CCR2 compared to HC (Fig. 2f). Collectively, this data shows that the largest variation occurred on IM where PLWH$_{ART}$ and PLWH$_{EC}$ showed a similar trend compared to HC.

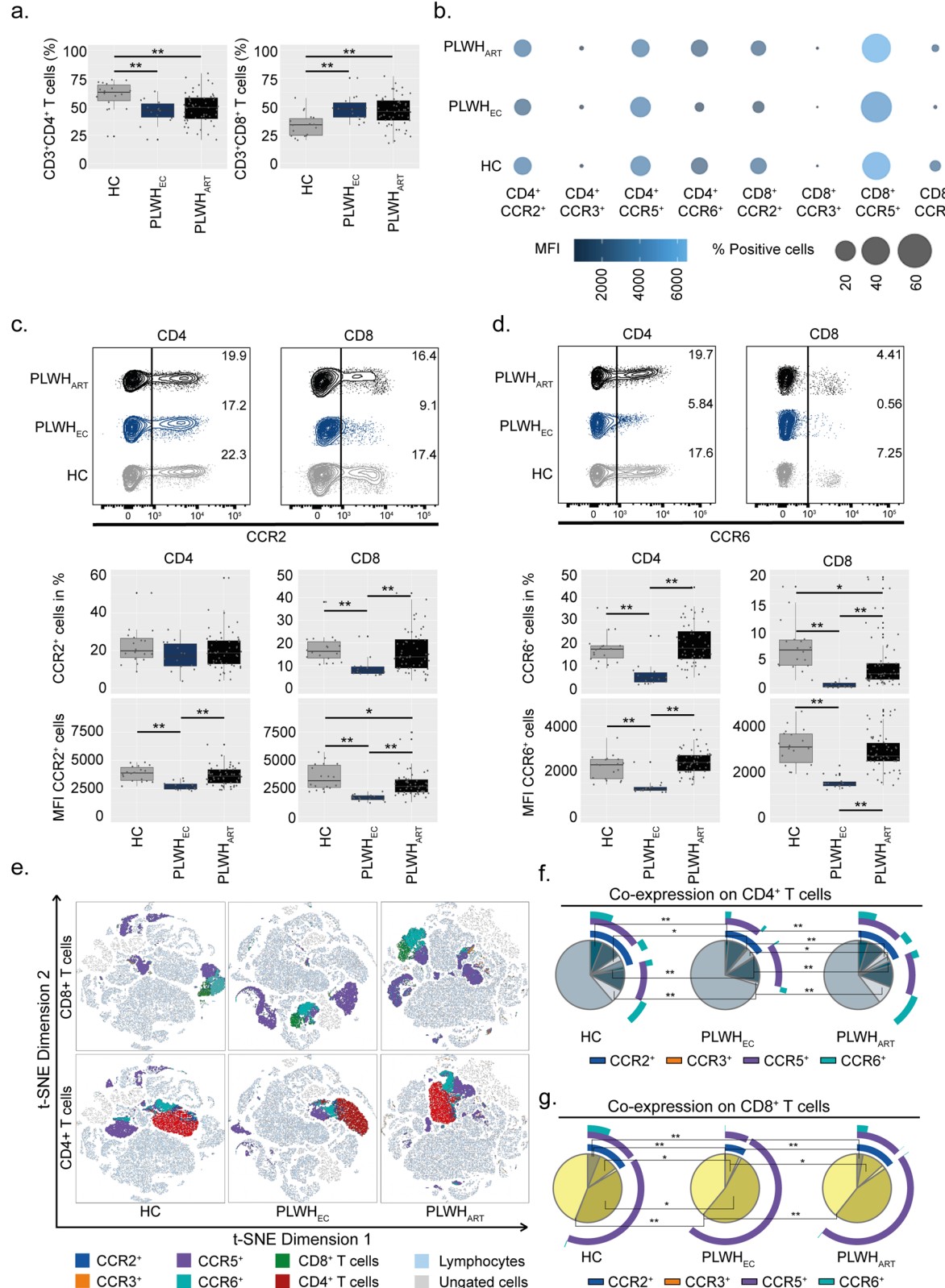

**Longer suppressive therapy reduces expression of CCR2 on monocytes.** The chronic inflammatory condition due to HIV-1 persistence and an activated immune response persists during suppressive ART. Therefore, to evaluate if longer treatment duration affects the receptor expression, we included longitudinal samples (timepoint 1 (T1) $n = 10$ and timepoint 2 (T2) $n = 10$) where the same patients were followed-up after 3 years of

additional suppressive therapy (clinical parameters can be viewed in Supplementary Table 2). Treatment duration did not affect frequencies of CD4$^+$, CD8$^+$ T cells or monocytic cell populations (Supplementary Fig. 4a, b). For the detected receptor expressions, most of the differences were detected in the monocytic cell populations while no significant variations were seen on CD4$^+$ T cells (Fig. 3a). Longer treatment duration increased the

**Fig. 1 CCR2 and CCR6 are specifically depleted from T lymphocytes in PLWH_EC.** Receptor expression on CD4$^+$ and CD8$^+$ T lymphocytes from PLWH_EC ($n = 14$), PLWH_ART ($n = 54$) and HC ($n = 18$). **a** Frequencies of CD4$^+$ T cells and CD8$^+$ T cells in total peripheral blood mononuclear cells derived from CD3$^+$ cells. **b** Bubble chart representing receptor expression of CCR2, CCR3, CCR5 and CCR6 on CD4$^+$ and CD8$^+$ T cells. Size of the bubble corresponds to percentage-positive cells while colour represents the median fluorescence intensity (MFI). **c** CCR2 receptor expression on CD4$^+$ and CD8$^+$ T cells (% cells and MFI). **d** CCR6 receptor expression on CD4$^+$ and CD8$^+$ T cells (% cells and MFI). **e** t-SNE plots showing receptor expression distribution on CD4$^+$ and CD8$^+$ T cells. **f** Co-receptor expression of CCR2, CCR3, CCR5 and CCR6 on CD4$^+$ T cells. **g** Co-receptor expression of CCR2, CCR3, CCR5, and CCR6 on CD8$^+$ T cells. **c**, **d** Contour plots show a representing sample from PLWH_EC, PLWH_ART and HC corresponding to the median % of cells within each group. **a**, **c**, **d**, **f**, **g** Statistical significance was determined using two-tailed Mann–Whitney U-test (significance level $p < 0.05$, with *<0.05, **<0.001) and represented as pie charts or with median using 95% CI.

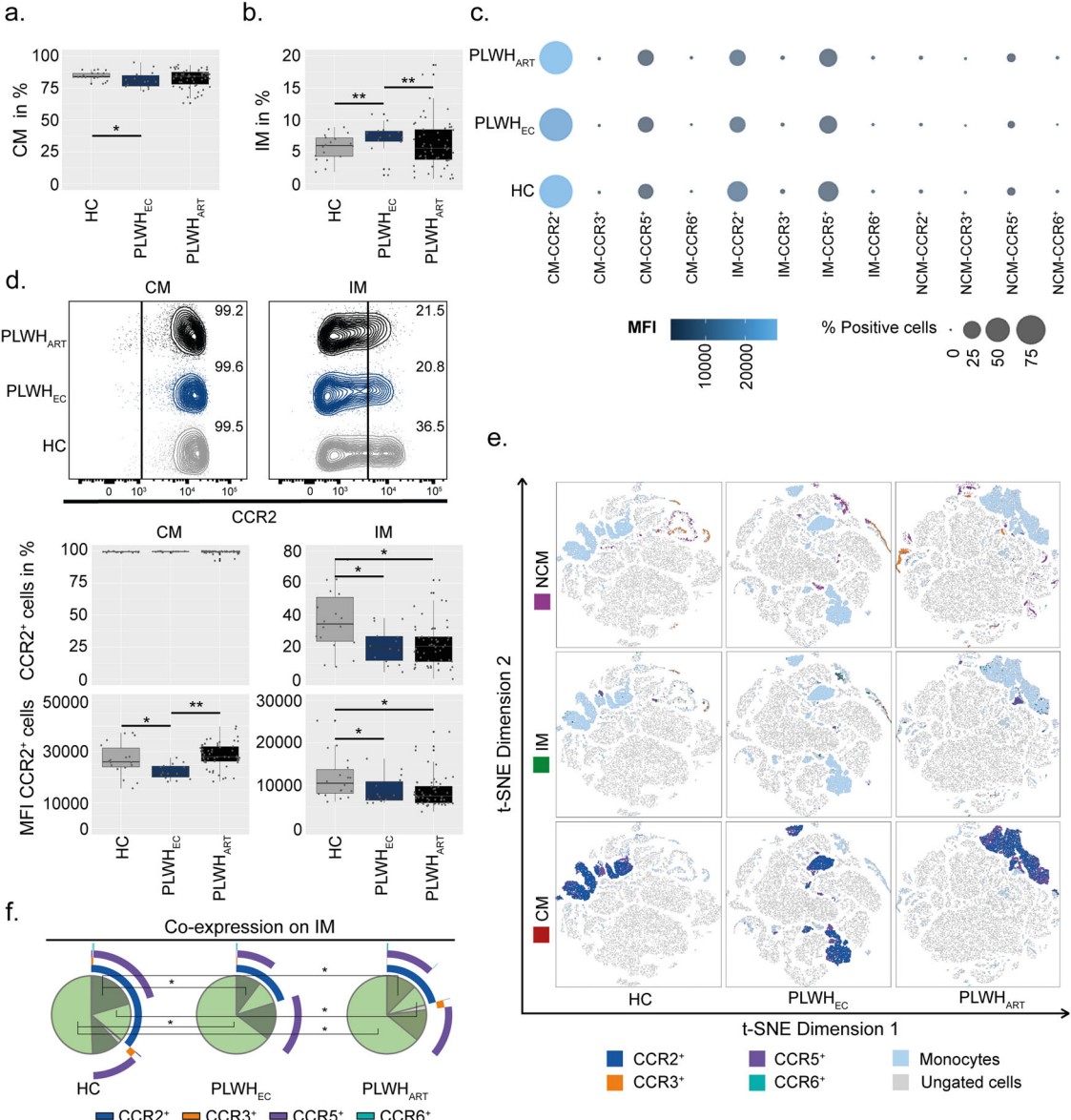

**Fig. 2 CCR2 is depleted from intermediate monocytes of PLWH.** Receptor expression on monocytic subpopulations (classical monocytes (CM); CD14$^+$CD16$^-$, intermediate monocytes (IM) CD14$^+$CD16$^+$ and non-classical monocytes (NCM); CD14$^-$CD16$^+$) in PLWH_EC ($n = 14$), PLWH_ART ($n = 54$) and HC ($n = 18$). **a**, **b** Percentages of CM (**a**) and IM (**b**) from CD3$^-$ cells in peripheral blood mononuclear cells. **c** Bubble chart representing receptor expression of CCR2, CCR3, CCR5 and CCR6 on CM, IM and NCM. Size of the bubble corresponds to percentage-positive cells while colour represents the median fluorescent intensity (MFI). **d** CCR2 receptor expression on CM and IM (% cells and MFI). **e** t-SNE plots showing receptor expression distribution of CCR2, CCR3, CCR5 and CCR6 on CM, IM and NCM. **f** Co-receptor expression of CCR2, CCR3, CCR5 and CCR6 on IM. **d** Contour plot shows a representing sample from PLWH_EC, PLWH_ART and HC corresponding to the median % of cells within each group. **a**, **b**, **d**, **f** Statistical significance was determined using two-tailed Mann–Whitney U-test (significance level $p < 0.05$, with *<0.05, **<0.001) and represented as pie charts or with median using 95% CI.

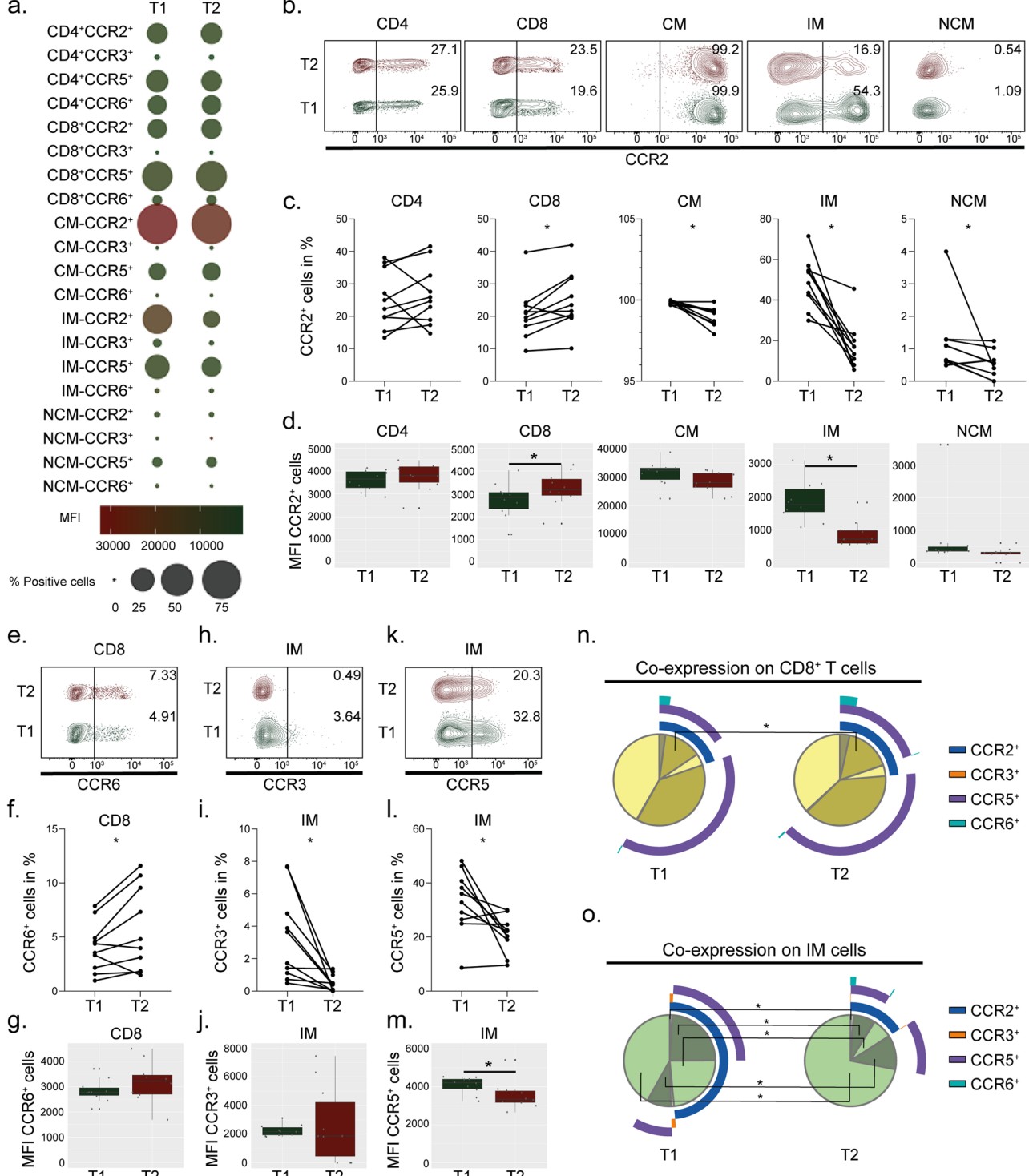

**Fig. 3 In time, CCR2 and CCR6 increase on CD8 cells while CCR2 decreases on monocytes in PLWH_ART.** Receptor expression of longitudinal data on lymphocytes and monocytes at arbitrary timepoint 1 (T1) ($n = 10$) and three years later at timepoint 2 (T2) ($n = 10$) from total peripheral blood mononuclear cells. **a** Bubble chart representing receptor expression of CCR2, CCR3, CCR5 and CCR6 on CD4+ and CD8+ T cells together with classical (CM; CD14+CD16−), intermediate (IM; CD14+CD16+), and non-classical (NCM; CD14−CD16+) monocytes. Size of the bubble corresponds to percentage-positive cells while colour represents the median fluorescent intensity (MFI). **b, c** Receptor expression of CCR2 on CD4+, CD8+, CM, IM and NCM. **d** MFI of CCR2 on CD4+, CD8+, CM, IM and NCM. **e, f** Receptor expression of CCR6 on CD8+ T cells. **g** MFI of CCR6 on CD8+ T cells. **h, i** Receptor expression of CCR3 on IM. **j** MFI of CCR3 on IM. **k, l** Receptor expression of CCR5 on IM. **m** MFI of CCR5 on IM. **n, o** Co-receptor expression of CCR2, CCR3, CCR5 and CCR6 on CD8+ T cells (**n**) and IM (**o**). **b, e, h, k** Contour plot shows a representing sample from T1 and T2 corresponding to the median % of cells within each group. **c, d, f, g, i, j, l–o** Statistical significance was determined using two-tailed Wilcoxon matched-pairs signed-rank test (significance level $p < 0.05$, with *<0.05, **<0.001) and represented as paired before and after plot, median using 95% CI or as pie charts.

frequency of CD8+CCR2+ cells, while all three monocytic cell populations exhibited reduced proportions of cells expressing CCR2 in T2 (Fig. 3b, c). Following a similar trend, CCR2 expression was increased on CD8+ T cells and IM exhibited reduced expression of CCR2 in T2 while CM and NCM showed no difference (Fig. 3d). The proportion of CD8+CCR6+ cells was increased in T2, without showing any significant effect on CCR6 expression levels (Fig. 3e–g). Furthermore, the proportions of IM-CCR3+ and IM-CCR5+ cells decreased over time while only expression levels of CCR5 decreased in T2 (Fig. 3h–m). Investigations of co-receptor expression on the lymphocytic cell populations showed that the only difference was an increase in CCR2+CCR5+ in T2 on CD8+ T cells (Fig. 3n). Within the monocytic cell populations, the only differences of co-expression of the receptors were seen on IM where T2 exhibited higher percentage of cells negative for all four receptors while CCR2+, CCR2+CCR3+ and CCR2+CCR5+ were decreased in T2 (Fig. 3o). Conclusively, the effect induced by longer suppressive treatment mainly reduces expression levels of these chemokine receptors on IM.

**The HIV-1 reservoir is lower in PLWH_EC compared to PLWH_ART.** To evaluate the proportion of HIV-1 infected cells in our cohort, we quantified total HIV-1 DNA in the PLWH_EC ($n = 14$) and PLWH_ART ($n = 54$). Herein, we observed a higher level of total HIV-1 DNA in PLWH_ART compared to PLWH_EC

(Fig. 4a) despite a median of eight years of successful treatment. No significant difference was observed between long-term ART (lnART; >10 yrs (median 20 yrs, Interquartile range (IQR); 18–22.75), $n = 20$) and short-term ART (sART; <10 yrs (median 7 yrs, IQR; 6–8), $n = 34$) nor was the reservoir affected by longer suppressive treatment between T1 and T2 within the same individuals (Fig. 4b, c). This data indicates that after the initial two phases of HIV DNA decline the size of the reservoir is not affected during successful suppressive ART. However, PLWH_EC exhibited reduced proportions of HIV-1 infected cells presumably due to their natural control of HIV-1. Next, to evaluate the proportion of integrated HIV-1 that was intact/defective, we employed the intact provirus DNA assay (IPDA)[28] together with quantification of the 5′LTR and 2-LTR circles on the genomic DNA from PBMCs. A significant increase of 3′deletion/hypermutation was detected in PLWH_ART compared to PLWH_EC, but no significant differences in intact provirus, 5′deletion, 5′LTR or 2-LTR circles between the two groups (Fig. 4d). Moreover, percentages of intact provirus out of total HIV-1 DNA did not differ between the groups (Fig. 4e). The CCL20 ligand for CCR6 can facilitate nuclear integration of HIV-1 by remodelling the actin cytoskeleton, but also directly inhibits infection in the female reproductive tract[29,30]. Therefore, to evaluate if the ligands for CCR2 and CCR6 might have an effect on the HIV-1 reservoir we performed correlation analysis between plasma levels of CCL20 CCL2 and the HIV-1 reservoir. No significant correlation between HIV-1 DNA and CCL20 plasma levels was revealed in

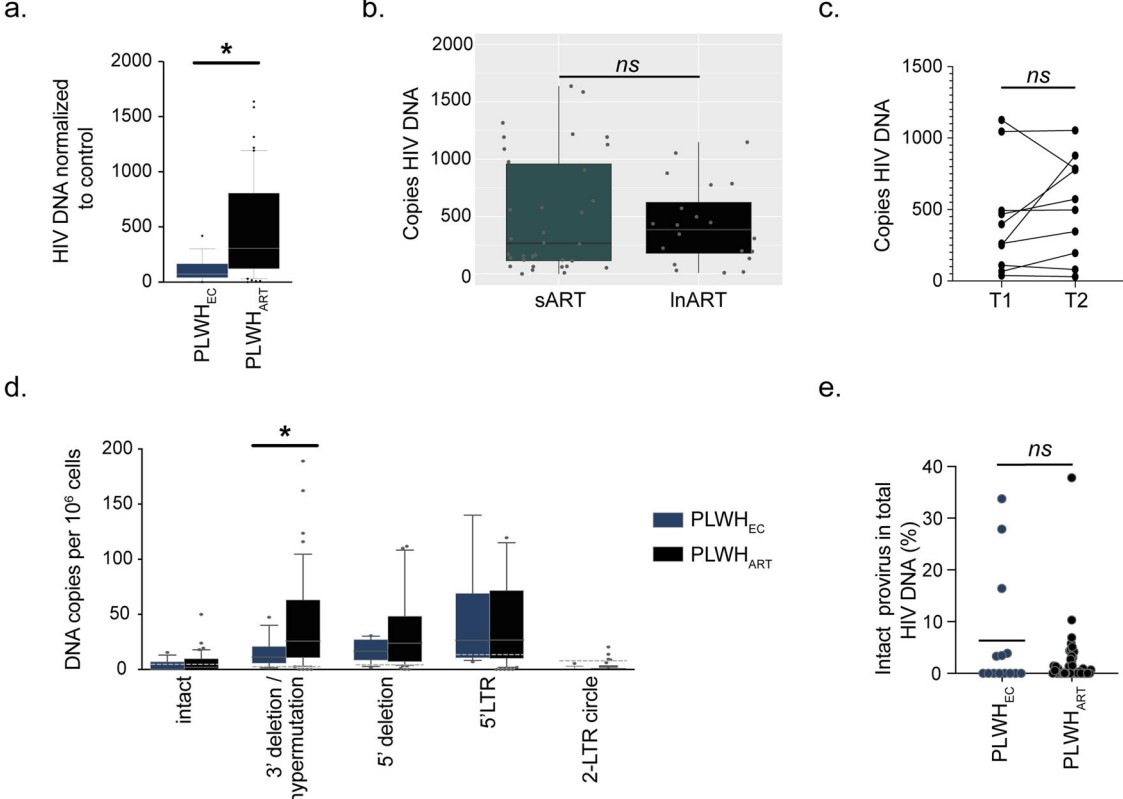

**Fig. 4 The HIV-1 reservoir in PLWH_EC is distinct from PLWH_ART. a** Quantified total HIV-1 DNA in peripheral blood mononuclear cells from PLWH_EC ($n = 14$) and PLWH_ART ($n = 54$). **b** Total HIV-1 DNA in short-term ART (sART, <10 yrs, $n = 34$) compared to long-term ART (lnART, >10 yrs, $n = 20$) in people living with HIV-1 on suppressive therapy. **c** Total HIV-1 DNA in longitudinal data from HIV-1 infected individuals (timepoint 1 (T1), $n = 10$ and timepoint 2 (T2), $n = 10$). **d** Box and Whisker plot show the distribution of integrated HIV-1 DNA copies per $10^6$ PBMCs in PLWH_ART ($n = 54$) and PLWH_EC ($n = 14$) with whiskers representing the 10–90 percentile. The median is represented by the line inside the box. The dotted line marks the background represented by HC samples. **e** Percentage of intact provirus out of total HIV-1 DNA detected. Statistical significance was determined using two-tailed Mann–Whitney U-test (significance level $p < 0.05$, with *<0.05, **<0.001) and represented as median using 95% CI. All individual samples were run in technical duplicates.

either PLWH$_{ART}$ or PLWH$_{EC}$ (Supplementary Fig. 5a). For CCL2, a negative correlation was detected in PLWH$_{EC}$ but not in PLWH$_{ART}$ (Supplementary Fig. 5b). Collectively, this data indicates that although PLWH$_{ART}$ have higher proportion of infected cells compared to PLWH$_{EC}$, a larger fraction of the integrated HIV-1 carries hypermutations or 3′deletions within the viral genome. Furthermore, the frequency of intact provirus out of total HIV-1 DNA is low in both PLWH$_{EC}$ and PLWH$_{ART}$.

**Altered metabolic pathways in CD4$^+$CCR6$^+$ cells are characterised by an upregulation in p53 signalling and apoptosis in PLWH$_{EC}$.** The subpopulation of CD4$^+$CCR6$^+$ cells has been described as more permissive towards HIV-1 and targets for productive infection[31,32]. To further characterise the CD4$^+$ T cells that express or lack the CCR6 receptor we sorted out these cell populations using fluorescence-activated cell sorting (FACS) from HC ($n = 3$), PLWH$_{EC}$ ($n = 6$) and PLWH$_{ART}$ ($n = 6$) (Fig. 5a). The PLWH$_{ART}$ samples were selected from PLWH on long-term suppressive ART (median 20 years, IQR; 17.5–22.25). First, as the HIV-1 reservoir has been proposed to reside in CD4$^+$CCR6$^+$ T cells, we measured the amount of integrated HIV-1 in PLWH$_{EC}$ ($n = 5$) and PLWH$_{ART}$ ($n = 3$). However, the variation within the data and the small sample size made the comparison inconclusive (Supplementary Fig. 6a). To evaluate the specific proteomic profile of CD4$^+$CCR6$^-$ and CD4$^+$CCR6$^+$ cell populations we isolated these cell populations by FACS (15,000–30,000 cells) and performed mass spectrometry-based proteomics using a quantitative tandem mass tags (TMT) labelling approach. Due to low protein detection by mass spectrometry four PLWH$_{EC}$ samples, two samples out of each of the CCR6$^+$ and CCR6$^-$ cell populations were excluded from the analysis. The principal component analysis (PCA) identified heterogeneity among the samples (Supplementary Fig. 6b). Differential expression analysis was unable to find statistically significant regulated proteins due to heterogeneity among samples and low sample size. Therefore, functional analysis was performed using a consensus scoring approach based on multiple protein set analysis (PSA) run by incorporating the directionality of protein abundance. Using the group-specific consensus scores (PLWH$_{EC}$ vs PLWH$_{ART}$) and directionality classes, we identified distinct upregulation of oxidative phosphorylation (OXPHOS) (adj $p < 0.05$) and interferon-α response (adj $p < 0.05$) and distinct downregulation of glycolysis (adj $p < 0.05$) in both CD4$^+$CCR6$^+$ and CD4$^+$CCR6$^-$ cells while using the MSigDB hallmark gene set (Fig. 5b). Similarly, the KEGG gene set also identified a distinct upregulation of OXPHOS (adj $p < 0.1$), and a distinct downregulation of other pathways of amino acid and carbohydrate metabolism (Supplementary Fig. 6c). This data indicates that the metabolic trade-off of increased OXPHOS and reduced glycolysis is elevated in PLWH$_{EC}$ compared to PLWH$_{ART}$ (Fig. 5c). The relative abundance of the OXPHOS proteins was higher in PLWH$_{EC}$ compared to HC while lower in PLWH$_{ART}$ compared to HC (Supplementary Fig. 7a, b). Interestingly, only p53 signalling and apoptosis were distinctly upregulated in CD4$^+$CCR6$^+$ (adj $p < 0.05$) (Fig. 5b) and both pathways had unique features in PLWH$_{EC}$ (Fig. 5d, e). The majority of the PLWH$_{ART}$ clustered together. There were two downregulated proteins in PLWH$_{EC}$ (ISCU and LDHB) that were common between OXPHOS and p53 signalling. Furthermore, four proteins (CASP7, VDAC2, RHOT2 and GPX4) were common between OXPHOS and apoptosis, while seven proteins (FAS, TAP1, FDXR, TXNIP, CASP1, APP and HMOX1) were common between p53 signalling and apoptosis among the proteins upregulated in PLWH$_{EC}$ compared to PLWH$_{ART}$ (Fig. 5f). Taken together, this data shows how the CD4$^+$CCR6$^+$ cell population have a unique profile in

PLWH$_{EC}$. The co-regulation of OXPHOS, p53 signalling and apoptosis may be a contributing factor to control HIV-1 infection.

**Discussion**
In our study, a unique cell type expression profile of CCR2 and CCR6 was identified in PLWH$_{EC}$, while PLWH$_{ART}$ exhibited an expression profile more like HC. CCR6$^+$ cells were thus decreased on both CD4$^+$ and CD8$^+$ T cells in PLWH$_{EC}$ but not in PLWH$_{ART}$. Furthermore, the CD4$^+$ T cells exhibited a protein profile of modulated metabolic activity towards OXPHOS, and reduced protein levels involved in glycolysis in PLWH$_{EC}$, irrespective of CCR6 status. Collectively, as CD4$^+$CCR6$^+$ cells have been reported to be highly permissive to HIV-1[31,32], and substantial contributors to the HIV-1 reservoir, low levels of CD4$^+$CCR6$^+$ cells, metabolic modulation, and enrichment of proteins involved in cell death, could potentially contribute to natural control of HIV-1.

Within the CD4$^+$ T cell subpopulation, CCR6 expression is a marker of some memory T cells subsets, namely Th17 cells, natural killer T (NKT) cells, and γδ T cells[33]. Of these cell populations, the Th17 cells promote a strong pro-inflammatory response and are drivers of chronic inflammation and autoimmune diseases[34,35]. Earlier studies have shown how memory CD4$^+$CCR6$^+$ T cells express a Th17 lineage profile and high susceptibility towards HIV-1[36]. In our cohort, the proportion of CCR6$^+$ cells was reduced in PLWH$_{EC}$ compared to both PLWH$_{ART}$ and HC. CCR6 is also a homing marker for gut-associated lymphoid tissue (GALT)[37]. Therefore, the causative factor of reduced CD4$^+$CCR6$^+$ cells in peripheral blood in PLWH$_{EC}$ could be tissue residency in GALT to a higher extent than in PLWH$_{ART}$. Although initiation of ART restores the intestinal CD4$^+$ T cell compartments, the severe depletion induced during the early stages of infection cannot be fully recovered[38], which contributes to exacerbation of HIV-1 through microbial translocation and hyperactivation of the immune system[39–41]. Collectively, an increased gut homing potential of CCR6$^+$ cells in PLWH$_{EC}$, could help to restore the imbalance of CD4$^+$ T cells in GALT and reduce systemic chronic inflammation among these patients.

We have earlier shown that low expression of CCR6 and high levels of its ligand, CCL20, may result in low levels of inflammation in PLWH$_{EC}$[25]. Herein, we detected increased plasma levels of CCL20 in PLWH, independent of HIV-1 status compared to HC, confirming earlier observations[42]. CCL20 has previously been described to have an effect on the infection steps of HIV-1 and migration/function of immune cells[33]. However, we did not see any difference in CCL20 levels between PLWH$_{EC}$ and PLWH$_{ART}$. Therefore, most likely it is the CD4$^+$CCR6$^+$ cells, rather than levels of CCL20, that in PLWH$_{EC}$ could contribute to natural control of HIV-1.

Furthermore, CD4$^+$CCR6$^+$ and CD8$^+$CCR6$^+$ cells decline in untreated patients with HIV-1[43] since the CCR6$^+$ cells are more susceptible towards apoptosis and therefore involved in the depletion of memory T cells. However, we did not detect any alterations in CD4$^+$CCR6$^+$ compartments in PLWH$_{ART}$ compared to HC, although for CD8$^+$CCR6$^+$ cells there was a reduction in both patient groups. We identified the CD4$^+$CCR6$^+$ cells as having increased proteins involved in apoptosis and p53 signalling in PLWH$_{EC}$, indicating an increased sensitivity towards apoptosis, possibly as a result of HIV-1 proteins, activation-induced cell death or induced expression of death receptors, e.g., Fas and TRAIL-RI/TRAIL-RII[44]. Since these cells are not restored in the GALT during ART, it can be postulated that the CD4$^+$CCR6$^+$ cell population is a major target for HIV-1

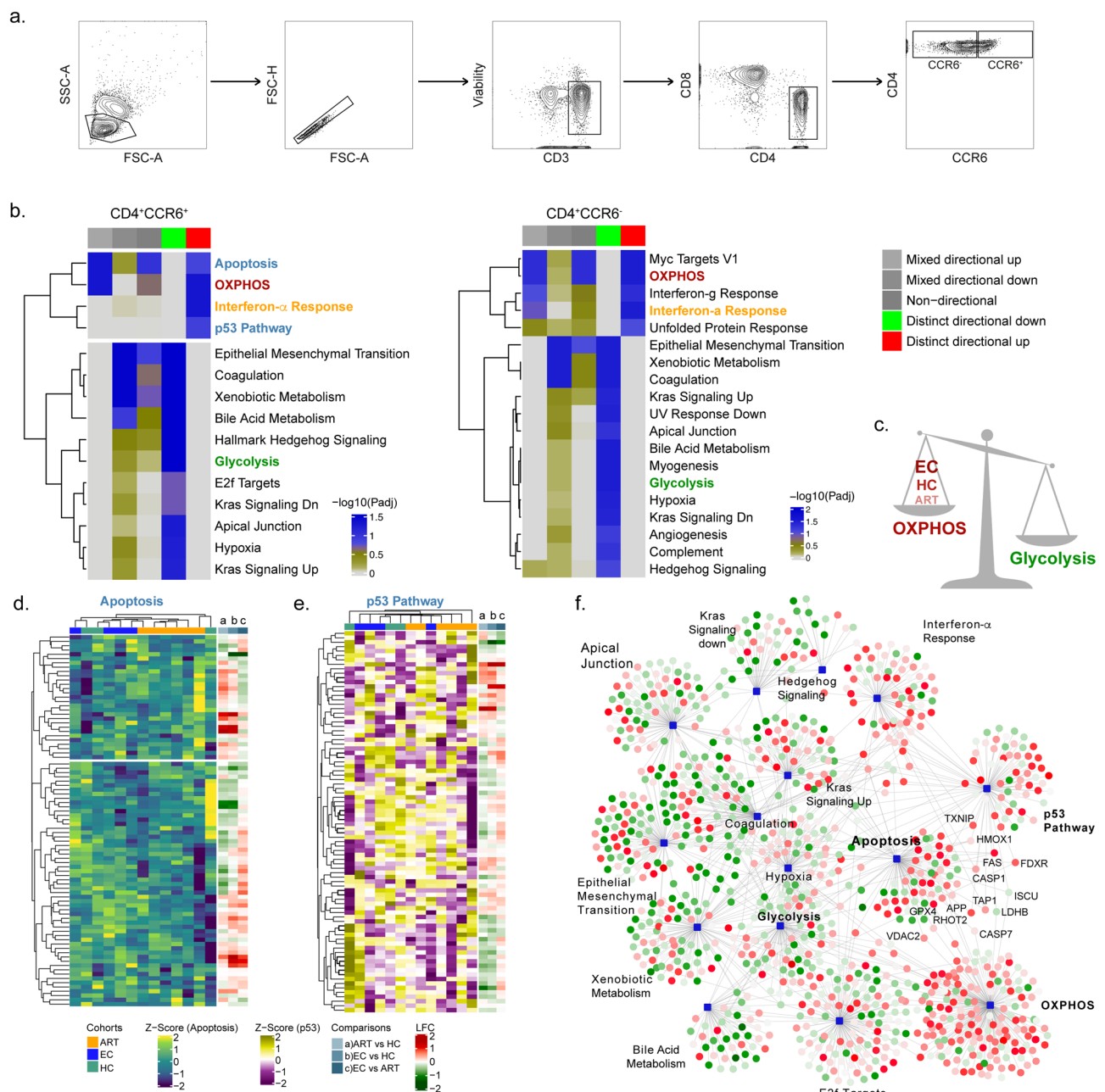

**Fig. 5 CD4⁺CCR6⁺ are enriched in proteins from p53 and apoptosis signalling in PLWH_EC compared to PLWH_ART.** Characteristics of CD4⁺CCR6⁺ and CD4⁺CCR6⁻ cell populations isolated by FACS from PLWH_EC ($n = 4$), PLWH_ART ($n = 6$) and HC ($n = 3$). **a** Gating strategy for isolation of CD4⁺CCR6⁺ and CD4⁺CCR6⁻ cell populations from total peripheral blood mononuclear cells, exhibited by an HC sample. **b** Heatmap representation of significant pathways found deregulated in CD4⁺CCR6⁺ PLWH_EC compared to CD4⁺CCR6⁺ PLWH_ART and in CD4⁺CCR6⁻ PLWH_EC compared to CD4⁺CCR6⁻ PLWH_ART using MSigDBv. Colour gradient corresponds to the negative log scaled adjusted p-values. Each column represents *p*-values of various directionality classes, calculated for the pathways. Non-directional p-values are calculated based on gene-level statistics regardless of the direction of expression. Mixed directional up and mixed directional down p-values are calculated using the subset of the gene statistics that are upregulated and downregulated, respectively. Distinct directional up and distinct directional down p-values are calculated from gene statistics with expression direction. **c** Schematic representation of metabolic trade-off between OXPHOS and glycolysis detected in the cells. **d, e** Heatmap representation of proteins part of apoptosis (**d**) and p53 pathway (**e**). Colour gradient corresponds to the z-score scaled normalized expression values. Column annotation represents the cohorts and row annotation visualizes the log2 transformed fold change values in each of the differential expression analysis of PLWH_ART compared with HC, PLWH_EC compared with HC and PLWH_EC compared with PLWH_ART. **f** Network visualization of significant pathways enriched in CD4⁺CCR6⁺ PLWH_EC compared to CD4⁺CCR6⁺ PLWH_ART. Each edge represents association of the protein with the corresponding pathways. Circular nodes denote proteins and square-shaped nodes are pathways. Red colour gradient and green colour gradient represent upregulation and downregulation of the proteins in CD4⁺CCR6⁺ PLWH_EC compared to CD4⁺CCR6⁺ PLWH_ART.

infection that in PLWH contributes to the inflammatory environment in both GALT and peripheral blood. In PLWH_EC, these cells may be more susceptible towards apoptosis, possibly through expression of pro-apoptotic HIV-1 proteins during infection or reactivation from latency. Furthermore, p53 is a tumour suppressor gene that during HIV-1 infection is activated by interferon α and β stimulation of immune cells[45]. Activity of p53 can interfere with HIV-1 replication by inhibiting reverse transcription of the virus and suppressing the activity of the transactivator tat, needed for viral transcription[46]. Therefore, an enrichment of p53 signalling in PLWH_EC could contribute to natural control of HIV-1 infection in CD4+CCR6+ T cells by reducing replication of HIV-1 and inducing apoptosis upon latency reversal. However, the sample size included in this study is relatively small and further studies validating these results are warranted. In addition, PLWH_EC exhibited an enrichment of proteins involved in OXPHOS together with decreased number of proteins involved in glycolysis, irrespective of CCR6 expression. Increased glycolysis is a hallmark for T cell activation but also upregulated during HIV-1 infection for cells to sustain the energy-demanding process of virus production[47,48]. Generally, activation of a T cell induces a shift from OXPHOS to aerobic glycolysis in a similar fashion as the metabolic reprogramming occurring during activation from latency[49,50]. Therefore, the metabolic state of the CD4+ T cell compartment in PLWH_EC, independently of CCR6 expression indicates a unique proportion of CD4 T cell subsets and metabolic reprogramming.

Persistent HIV-1 is a driver of chronic immune activation and inflammation. In our cohort, PLWH_EC had a reduced amount of integrated HIV-1 in total PBMCs compared to PLWH_ART, correlating to earlier observations[23]. On the other hand, treatment duration did not significantly alter the proportion of latently infected cells probably due to the stability of the reservoir during long-term ART. During suppressive therapy, persisting HIV-1 contributes to low levels of viral replication and cell-to-cell spread driving the chronic immune cell activation and inflammaging that ultimately sustains the HIV-1 reservoir in the body, together with homoeostatic proliferation[51,52]. However, evaluation of the intact provirus by IPDA showed a decrease of 3′deletion/hypermutation in the provirus from PLWH_EC compared to PLWH_ART. Earlier studies have shown that 3′ deletion clones decline during suppressive therapy[53]. These specific mutated proviruses may be under negative selection due to their capacity to stimulate the immune system[53,54]. With a reduced proportion of provirus in PLWH_EC compared to PLWH_ART, it could be hypothesised that this negative selection is stronger in PLWH_EC, explaining the lower proportion of integrated HIV-1.

Monocytes produce pro-inflammatory cytokines where IM are efficient producers of IL6 and IL8 in response to microbial pathogens together with CM, whereas NCM produce TNFα, CCL3 and IL-1β in response to viral elements[55]. In our cohort, elevated levels of IM were detected in PLWH_EC, indicative of an increased pro-inflammatory response. However, the expression profile of CCR2 on IM was similar between PLWH_EC and PLWH_ART. Furthermore, expression of CCR2 decreased on IM during longer suppressive therapy in our longitudinal PLWH_ART cohort and a similar trend was seen in PLWH_EC and PLWH_ART compared to HC. Monocytes are targets of HIV-1 infection and drivers of inflammation and comorbidities during HIV-1 infection[56]. Therefore, the expansion of IM could indicate a role in a heightened pathogen defence in PLWH_EC.

Overall, our study showed that the chemokine receptor profile in PLWH_ART is more similar towards HC on lymphocytic cell populations, and PLWH_EC on monocytic cell populations. PLWH_EC have a reduced proportion of CD4+CCR6+ cells in peripheral blood compared to PLWH_ART. Based on our proof of principle proteomic analysis we hypothesise that the CD4+CCR6+ cells are more susceptible to apoptosis, while CD4+ T cells in PLWH_EC (CCR6−/CCR6+) exhibit decreased metabolic activation, indicative of a reduced activation state. Collectively, these lymphocytic cell populations may aid in facilitating the unique characteristics of PLWH_EC and contribute to chronic inflammation and HIV-1 persistence during suppressive therapy.

## Methods

**Patient material**. The clinical cohort consisted of HIV-1+ individuals on suppressive therapy (PLWH_ART, n = 54), elite controllers (PLWH_EC, n = 14) collected from the InfCareHIV cohort at the Department of Infectious Diseases, Karolinska University Hospital, Sweden, and HIV-1 negative individuals (HC, n = 18). The cohort was age-matched and the median duration of treatment was 8 years (IQR: 6.75–19). Cohort characteristics can be viewed in Table 1 and treatment regimen for PLWH_ART in Supplementary Table 1. The cohort also consisted of longitudinal data within the PLWH_ART group (T1, n = 10 and T2, n = 10) (Supplementary Table 2). At the time of sampling, none of the study participants had any co-infections. Patient material consisted of plasma and PBMCs collected from whole blood using Ficoll-Paque (Cytiva).

This study was approved by the Regional Ethics Committee of Stockholm. Informed consent was given from all participants prior to inclusion, and data were anonymized and delinked before analysis.

**Flow cytometry and FACS**. PBMCs were stained for cell surface markers for 20 min at room temperature (RT) using 5 μL/10^6 PBMCs of CD3 (OKT3, FITC, Biolegend #317306), CD4 (SK3, BUV395, BDbioscience #563550), CD8 (RPA-T8, APC, Biolegend #301014), CD14 (M5E2, BV510, Biolegend #301842), CD16 (3G8, BV786, BDbioscience #563690), CCR2 (1D9, BB700, BDbioscience #747847), CCR3 (5E8, BV421, Biolegend #310714), CCR5 (2D7, PE-CF594, BDbioscience #562456), CCR6 (G034E3, BV711, Biolegend #353436), and Near IR viability dye (Invitrogen #L10119). All antibodies were used at a volume of 5 μL/100 μL reaction and Near IR viability dye at a concentration of 1:100. Cells were subsequently washed two times using FACS buffer (PBS with 2% FBS and 2 mM EDTA) and fixed 15 min at RT using 2% paraformaldehyde. Results were acquired on BD FACS Symphony (BD Bioscience, USA) using laser and filter settings as indicated for BUV395, BV421, BV510, BV711, BV786, FITC, BB700, PE-CF594, APC and Near IR, respectively; 355 nm UV (100 mW) 379/28, 405 nm violet (100 mW), 450/50, 525/50 (505 LP), 710/50 (685 LP), 810/40 (770 LP), 488 nm blue (200 mW) 530/30 (505 LP), 710/50 (685 LP), 561 nm Y/G (200 mW), 610/20 (600LP) and 637 nm Red (140 mW), 670/30 and 780/60 (750LP). Cell expression analysis and t-SNE were performed using FlowJo 10.6 (Tree Star Inc). Co-expression analysis was performed using the Boolean gating function and the visualisation of complex data using Spice[57]. Data from flow cytometry detection in PLWH_EC (n = 14) and HC (n = 8) samples were used from our earlier publication[25].

Cryopreserved PBMCs were used for fluorescent activated cell sorting (FACS) for subsequent proteomic analysis and DNA extraction. Cells were thawed and stained with anti-human CD3 (OKT3, FITC, Biolegend #317306), CD4 (SK3, PE-Cy5, Biolegend # 344654), CD8 (RPA-T8, BV570, Biolegend #301038), CD14 (M5E2, BV510, Biolegend #301842), CD16 (3G8, BV786, BDbioscience #563690), CCR6 (G034E3, BV711, Biolegend #353436) for 30 min at 4 °C and subsequent FVS450 viability dye (BDbioscience #562247) for 10 min at RT. All antibodies were used at a volume of 5 μL/100 μL reaction and FVS450 was used at 1:1000. CD4+CCR6− and CD4+CCR6+ cell was collected for downstream analysis. Cells were sorted using SONY MA900 (SONY cooperation, Tokyo, Japan).

**DNA extraction**. Total DNA was extracted from whole PBMCs or collected cell populations by QIAamp Mini or QIAamp Micro DNA isolation kit (Qiagen), according to the manufacturers protocol.

**HIV-1 DNA quantification**. Patient PBMCs or sorted cells were used for quantification of total HIV-1 DNA by Internally Controlled qPCR (IC-qPCR) developed by Vicenti et al.[58]. In brief, total HIV-1 DNA was quantified using Takara Universal mastermix on the ABI 7500F and normalized to β-Actin levels, primer and probe details are described in Supplementary Table 3. For each sample, 200–500 ng were run per reaction in duplicates and data were normalised to background detection in HC.

**Digital droplet PCR and intact proviral DNA assay (IPDA)**. Digital droplet PCR (ddPCR) was performed using the QX200 Droplet Digital qPCR System (RioRad). Samples consisting of 10 μL Supermix for probes (no dUTPs) (Bio-Rad), 900 nM primers, 250 nM probe (labelled with either HEX or FAM) or 500 ng of genomic sample DNA and 50 ng of DNA for RPP30 quantification for shearing analysis and normalization, were emulsified with the QX200 droplet generator. PCR reactions were subsequently performed on a C1000 thermal cycler (Bio-Rad) using the following thermal cycling protocol: 10 min at 95 °C for enzyme activation followed by 45 annealing/extension cycles of 30 s at 94 °C and 60 s at 57 °C and a final enzyme

deactivation step at 98 °C for 10 min. Subsequently, samples were read with the QX200 Droplet Reader (Bio-Rad) and analysed with the QuantaSoft software version 1.5 (Bio-Rad).

For the quantification of intact, 5′ deleted, and 3′ deleted/hypermutated proviruses, the Intact Proviral DNA Assay was performed as previously described[28]. Briefly, a first ddPCR was performed as described above with primer/probe combinations for the packaging signal (Ψ), which frequently shows small mutations. The second reaction targets the HIV-1 Rev response element, present in the Env coding region. Two hydrolysis probes were used in this reaction: a HEX-labelled probe specific for the intact proviral sequence and an unlabelled probe specific for the APOBEC-3G hypermutation, which competes with the HEX-labelled probe and results in no signal in the case of APOBEC hypermutations. Duplicate wells for IPDA were merged during analysis. All primer and probe details can be viewed in Supplementary Table 3.

**Targeted proteomic profiling**. Plasma levels of CCL20 were measured using Human Quantikine ELISA kits (R&D Systems) according to the manufacturers protocol. Data for plasma CCL2 was used from our earlier publication[18]. In brief, plasma levels were analysed using proximity extension assay with the OLINK® immune-oncology panel (Olink, Sweden). Proteins are reported as normalised protein expression levels (NPX) where Ct values are normalised by the subtraction of the extension control and inter-plate control.

**Mass spectrometry-based proteomics**. Proteins were extracted by adding 8 M urea in 100 mM ammonium bicarbonate lysis buffer. Samples were vortexed and sonicated in cold bath for 10 min. Protein concentration was estimated with Pierce Micro BCA assay (ThermoFisher Scientific). Extracted proteins were reduced with 10 mM dithiothreitol for 60 min at 37 °C and alkylated with 40 mM chloroacetamide (CAM) for 30 min at room temperature. Proteins were digested with trypsin (sequencing grade from Promega) with a 1:20 ratio enzyme:protein overnight at 37 °C. Resulting peptides were cleaned-up using C-18 stage tips (ThermoFisher Scientific) according to manufacturer's instructions and dried in a vacuum concentrator (Eppendorf). Cleaned peptides were labelled with TMT11plex™ tags (ThermoFisher Scientific) following manufacturer's instructions. Briefly, peptides were resuspended in 100 mM triethylammonium bicarbonate and individual TMT tags were dissolved in dry acetonitrile (ACN). Peptides were labelled by adding the respective TMT tag solution, the mix was incubated for 1 h at room temperature, the reaction was quenched by adding 5% hydroxylamine and incubated for 15 min at RT. Labelled peptides were mixed in three different batches according to the scheme described in Supplementary Table 4, a pool of samples was used across the three batches to normalize and evaluate batch effect (TMT channel 131C).

Labelled peptides were fractionated using Pierce High pH Reversed-Phase Peptide Fractionation Kit columns (ThermoFisher Scientific) following manufacturer's instructions. Briefly, columns were washed with 100% ACN and then conditioned with 0.1% trifluoroacetic acid (TFA). Peptides were dissolved in 0.1% TFA and loaded into the column by centrifugation at $3000 \times g$ for 2 min. Columns were washed with 5% ACN in 0.1% triethanolamine (TEA), and peptides were eluted in eight different fractions by centrifugation at $3000 \times g$ according to the ACN concentration in 0.1% TEA solutions: 10, 12.5, 15, 17.5, 20, 22.5, 25 and 50%. All fractions were dried down in a vacuum concentrator (Eppendorf) prior to LC-MS/MS analysis.

Fractionated peptides were analysed on an Ultimate 3000 UHPLC (ThermoFisher Scientific) hyphenated to an Orbitrap™ Lumos™ mass spectrometer (ThermoFisher Scientific). Peptides were loaded in an Acclaim PepMap trap column, 2 μm × 75 μm ID × 2 cm (Thermo Scientific) and separated in an EASY-Spray™ HPLC Column, 2 μm × 75 μm ID × 500 mm (Thermo Scientific) using a 120 min linear gradient. Data were acquired in data-dependent acquisition (DDA) mode, isolating the top 20 most intense precursors at 120,000 mass resolution in the mass range of $m/z$ 375–1400, maximum injection time (IT) of 50 ms and dynamic exclusion of 30 s, precursors were isolated with 0.7 Th width. MS/MS scans were obtained using high collision energy of 34%, resolution of 45,000 and maximum IT of 86 ms.

Proteins were searched against both SwissProt human and HIV-1 databases using the search engine Mascot Server v2.5.1 (MatrixScience Ltd) in Proteome Discoverer v2.5 (ThermoFisher Scientific) software environment allowing maximum two missed cleavages. Oxidation of methionine, deamidation of asparagine and glutamine, TMT6plex modification of lysine and N-termini were set as variable modifications, while carbamidomethylation of cysteine was used as fixed modification. The false discovery rate (FDR) was set to 1%.

**Bioinformatics analysis**. The data were normalized using quantile method from the R/Bioconductor package NormalyzerDE v1.4.0. Missing data were imputed by employing the nearest-neighbour averaging method using impute.knn function from R package impute v1.60.0. Default setting was used for the data imputation. Technical variations in the data due to batch effect were removed using R function Combat from the package sva v3.34.0. Differential expressions analysis was performed with R/Bioconductor package Limma v3.42.2. Gene-set enrichment analysis was performed using R/Bioconductor package piano v2.2.0. Gene-level

t-statistics given by Limma, and hallmark gene set downloaded from MSigDB were used to find significantly enriched gene-sets. Gene-sets with Benjamini–Hochberg adjusted $p$-values <0.2 were considered as significantly enriched. Volcano and bubble plots were created using R package ggplot2 v3.3.2.

**Statistics and reproducibility**. Statistical analysis was performed using two-tailed Mann–Whitney U-test, and Wilcoxon ranked sum test or paired $t$-test for longitudinal data, based on the distribution of the data in Prism v8 (GraphPad Prism Software). Statistical significance was $p < 0.05$ for all analysis. Data were visualized using the package ggplot2 v.3.3.2 in Rstudio (v.1.3.1056). Correlation analysis was performed using Spearman's correlation ($p < 0.05$) in Rstudio (v.1.3.1056). All assays were performed in technical duplicates except for the flow cytometry evaluation that was performed in one replicate per sample.

**Reporting summary**. Further information on research design is available in the Nature Research Reporting Summary linked to this article.

## Data availability

The mass spectrometry proteomics data have been deposited to the ProteomeXchange Consortium via the PRIDE partner repository with the dataset identifier PXD027749. Raw data for FACS and flow cytometry are available from the corresponding authors upon request. The source data underlying most graphs used in this manuscript are provided as a Supplementary data file (Supplementary Data 1, excel).

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

## Acknowledgements

This study was funded by the Swedish Research Council Grants (2017-01330 to U.N. and 2017-05848 to A.S.) and Karolinska Institutet Stiftelser och Fonder. U.N. was also supported by the Swedish Research Council Interdisciplinary Grant (2018-06156). We would like to thank Assistant professor Robert van Domselaar for providing intellectual input on immunological aspects of the manuscript. Furthermore, we would like to acknowledge Dr. Sivasankaran Munusamy Ponnan for performing the t-SNE plots.

## Author contributions

U.N. conceived the study; A.S. initiated, designed, and governed the Swedish Elite controller cohort; U.N. and S.S.A. planned the experiments; S.S.A., S.K., S.G., B.J., and M.S. performed the research; J.E.R. and A.V. performed the proteomic analysis; A.T.A. performed the bioinformatics analysis. A.S., P.N. and J.V contributed with patient material. S.S.A. and U.N. analysed the data. S.G. and P.S. contributed with intellectual input. S.S.A. and U.N. wrote the manuscript. All authors critically reviewed and revised the manuscript.

## Funding

## Competing interests

The authors declare no competing interests.
