## [Peer Review File · Communications Biology]

Reviewers' comments:

Reviewer #1 (Remarks to the Author):

In this manuscript, Akusjärvi et al, present an assessment of chemokine receptor expression dynamics on various immune cell populations in people living with HIV (PLWH) that have been identified as elite controllers (PLWHEC) as compared to PLWH on suppressive antiretroviral therapy (PLWHART) and healthy controls (HC). The authors hypothesize that chemokine receptor dynamics in PLWHEC as compared to PLWHART may differentiate the two HIV-infected populations and provide a potential mechanism underlying the ability of PLWHEC to better manage HIV infection without the aid of ART. To test this hypothesis, the authors use flow cytometry to characterize the frequency and expression level of various chemokine receptors on CD4 and CD8 T cells, as well as classical, intermediate, and non-classical monocytes, among total PBMCs from PLWHEC, PLWHART, and HCs. The authors demonstrate that PLWHEC exhibit alterations in single and co-expression of CCR2, CCR3, CCR5 and CCR6 among T cells and monocyte subpopulations, as compared to PLWHART and HCs. The authors further perform a longitudinal analysis and demonstrate that a greater amount of time on ART only significantly impacted chemokine receptor expression on intermediate monocytes. The authors additionally assess differences in the HIV reservoir and demonstrate differences in the proportion of infected cells, including the fraction of cells with intact proviruses, between PLWHEC and PLWHART. Finally, the authors perform proteomics analysis of purified CCR6+ and CCR6- CD4+ T cells and present data suggesting that CCR6 expressing CD4 T cells may be skewed towards a more apoptotic profile. The authors conclude that PLWHEC differ from PLWHART in terms of the frequency and phenotype of CCR6 expressing CD4+ T cells, which may contribute to the increased ability of PLWHEC to better control inflammation and viral reservoir as compared to PLWHART. The results are carefully presented, however there are additional analyses and details which would improve the manuscript. Specific comments are listed below:

- Abstract, page 1: The abstract focuses mainly on the metabolic pathway findings, which is only a small portion of the data presented in the manuscript. Very little description is included of the different cell subsets that are examined, including both T cells and monocyte populations, the frequency of these subsets that express the chemokine receptors under consideration, the level of expression of chemokine receptors on each cell subset, and the effects of longer suppressive therapy on the frequency of chemokine receptor expressing subsets and level of expression. The abstract should be updated to more fully reflect all the findings presented in the manuscript.
- Abstract, page 1, Lines 26-27: "As of today, the effect of viral suppression and immune reconstitution in people living with HIV-1 (PLWH) on long-term suppressive ART (PLWHART) is not well-described". This sentence is a bit disingenuous, as the incomplete resolution of inflammation and immune activation in PLWH despite long term ART has been well documented. This should be restated for clarity.
- The presence of co-infections, like viral hepatitis or other sexually transmitted infections, are common among PLWH and can influence inflammation and immune activation. Were any of the study participants know to have other co-infections that could confound the results and if so, were these co-infections adjusted for in the statistical analyses?
- Relatedly, were statistical adjustments for multiple comparisons and other confounding factors, like age, sex, etc., included in the analyses?
- Only a very small subset of participant samples was included in the proteomics analysis. Is there any way for the authors to confirm the findings that apoptotic profiles may differ between PLWHEC and PLWHART, for example through flow cytometry staining for active caspase 3 expression or similar by CCR6+ and CCR6- CD4 T cells?
- Figure 1b, Figure 2c, Figure 3a: The bubble plots are difficult to interpret, and it would be more helpful to be able to see the individual data points for each study participant, as well as error bars, similarly to how the data is shown in panel 1a.
- Figures 1, 2, 3, and 5: Are the contour plots included in these figures from a PLWHEC, PLWHART, or HC? This should be specified in the figure legend.
- Figure 5: The number of samples in each study participant group is missing from the figure legend.
- Several of the panels in the supplementary materials should be moved to the main text. In particular, the authors should consider moving Supplementary Figures 2a and b, Supplementary

Figure 3b, and Supplementary Figure 5a, b and c to the main text.

- Lines 172-173: There appears to be a typo as the sentence should likely be “significant effect of CCR2 expression levels”, not CCR6.
- Lines 271-273: This sentence appears to be missing a reference.

Reviewer #2 (Remarks to the Author):

The article by Akusjarvi et al investigates chemokine receptor expression on T cell and monocyte subsets from peripheral blood mononuclear cells in chronically infected HIV patients with viral loads below the detection level (<40 copies/mL). Patients were either elite controllers (EC; n = 14) or on anti-retroviral therapy (ART; n=54). A cohort of n =18, HIV negative individuals were also studied. The authors investigated the expression levels of CCR2, CCR3, CCR5, and CCR6 in T cells and monocytes by flow cytometry and performed latent reservoir analysis on bulk CD4 T cells. The data while interesting, are not presented in a logical cohesive manner and appear to be extremely fragmented. This in addition to the lack of sufficient rigor significantly dampens enthusiasm for the study.

1. The flow of the Abstract is confusing as the first few lines imply an analysis focused on understanding the inflammatory consequences of HIV infection despite viral suppression. But the remainder of the Abstract then focusses on the phenotype of CD4 T cells in ECs.

2. Figure 1. The authors should clarify the parent gate that CD4 and CD8 T cells derived from - % of what subset? The total appears to be greater than 100.

CD4 T cells should be sub-setted by naive and memory subsets

The authors show MFI differences in CCR2 and CCR6 – since MFI is highly dependent on laser intensity and essentially the intensity can vary from day-to-day, what measures were taken to ensure consistency? Was an internal control used?

Enumerating frequency of CCR2 and CCR6+ cells would be informative.

3. Figure 2. Similar to expression profile in T cells, was the frequency of CCR2- expressing monocytes altered in HIV infection? Quantifying the ligands for CCR2 and CCR6 would have been informative.

4. Figure 3 shows longitudinal increase in MFI and frequencies of CCR2 and CCR6 on CD8 T cells after 3 years. What was the viral load data at T1 and T2?

5. Measurement of the HIV reservoir across EC and ART suppressed patients could have been performed on CCR6+ versus CCR6- CD4 T cells.

Response to Reviewers:

Dear Editor,

We are thankful to the reviewers for their constructive criticism and appreciate their time for the review. We have now responded point by point, and accepted all the suggestions. The changes in the manuscript are marked in red text.

Reviewers' comments:

Reviewer #1 (Remarks to the Author):

In this manuscript, Akusjärvi et al, present an assessment of chemokine receptor expression dynamics on various immune cell populations in people living with HIV (PLWH) that have been identified as elite controllers (PLWHEC) as compared to PLWH on suppressive antiretroviral therapy (PLWHART) and healthy controls (HC). The authors hypothesize that chemokine receptor dynamics in PLWHEC as compared to PLWHART may differentiate the two HIV-infected populations and provide a potential mechanism underlying the ability of PLWHEC to better manage HIV infection without the aid of ART. To test this hypothesis, the authors use flow cytometry to characterize the frequency and expression level of various chemokine receptors on CD4 and CD8 T cells, as well as classical, intermediate, and non-classical monocytes, among total PBMCs from PLWHEC, PLWHART, and HCs. The authors demonstrate that PLWHEC exhibit alterations in single and co-expression of CCR2, CCR3, CCR5 and CCR6 among T cells and monocyte subpopulations, as compared to PLWHART and HCs. The authors further perform a longitudinal analysis and demonstrate that a greater amount of time on ART only significantly impacted chemokine receptor expression on intermediate monocytes. The authors additionally assess differences in the HIV reservoir and demonstrate differences in the proportion of infected cells, including the fraction of cells with intact proviruses, between PLWHEC and PLWHART. Finally, the authors perform proteomics analysis of purified CCR6+ and CCR6- CD4+ T cells and present data suggesting that CCR6 expressing CD4 T cells may be skewed towards a more apoptotic profile. The authors conclude that PLWHEC differ from PLWHART in terms of the frequency and phenotype of CCR6 expressing CD4+ T cells, which may contribute to the increased ability of PLWHEC to better control inflammation and viral reservoir as compared to PLWHART. The results are carefully presented, however there are additional analyses and details which would improve the manuscript. Specific comments are listed below:

1. Abstract, page 1: The abstract focuses mainly on the metabolic pathway findings, which is only a small portion of the data presented in the manuscript. Very little description is included of the different cell subsets that are examined, including both T cells and monocyte populations, the frequency of these subsets that express the chemokine receptors under consideration, the level of expression of chemokine receptors on each cell subset, and the effects of longer suppressive therapy on the frequency of chemokine receptor expressing subsets and level of expression. The abstract should be updated to more fully reflect all the findings presented in the manuscript.

Response: We would like to thank the reviewer for pointing out this important aspect. We agree that the abstract was only focused on a small part of the study and have now tried to address this to give a more broader and complete summary of the study. To fully reflect the content of the manuscript we have rewriting it to say: "Herein, we show how PLWH who naturally control the virus (PLWH_{EC}) have a reduced proportion of CD4⁺CCR6⁺ and CD8⁺CCR6⁺ cells compared to PLWH on suppressive ART (PLWH_{ART}) and HIV-1 negative controls (HC). Expression of CCR2 was reduced on both CD4⁺, CD8⁺, CM, and IM monocytes in PLWH_{EC} compared to PLWH_{ART} and HC. Longer suppressive therapy, measured in the same patients, decreased number of cells expressing CCR2 on all monocytic cell populations while expression on CD8⁺ T cells increased. Furthermore, the CD4⁺CCR6⁺/CCR6⁻ cells exhibited a unique proteomic profile with a modulated energy metabolism in PLWH_{EC} compared to PLWH_{ART} independent of CCR6 status. The

Response to Reviewers:

CD4⁺CCR6⁺ cells also showed an enrichment in proteins involved in apoptosis and p53 signalling in PLWH_{EC} compared to PLWH_{ART}, indicative of increased sensitivity towards cell death mechanisms. Collectively, this data shows how PLWH_{EC} have a unique chemokine receptor profile that may aid in facilitating natural control of HIV-1 infection" on line 33-44.

2. Abstract, page 1, Lines 26-27: "As of today, the effect of viral suppression and immune reconstitution in people living with HIV-1 (PLWH) on long-term suppressive ART (PLWHART) is not well-described". This sentence is a bit disingenuous, as the incomplete resolution of inflammation and immune activation in PLWH despite long term ART has been well documented. This should be restated for clarity.

Response: We appreciate that the reviewer has pointed this out to us and have now tried our best to clarify the sentence. The updated version now says: "As of today, the effect of viral suppression and immune reconstitution in people living with HIV-1 (PLWH) has been well described but not completely understood" on line 32-33.

3. The presence of co-infections, like viral hepatitis or other sexually transmitted infections, are common among PLWH and can influence inflammation and immune activation. Were any of the study participants know to have other co-infections that could confound the results and if so, were these co-infections adjusted for in the statistical analyses?

Response: We thank the reviewer for this reply and agree that this is defiantly an important factor to consider. To minimize any confounding effects in our cohort we only included study participants who did not have any known co-infections at the time of sampling, and during treatment timepoints for the longitudinal cohort. We have now added this sentence in the manuscript as follows: "At time of sampling, none of the study participants had any co-infections" on line 108 in the material and methods.

4. Relatedly, were statistical adjustments for multiple comparisons and other confounding factors, like age, sex, etc., included in the analyses?

Response: This is an important point that we are thankful that the reviewer pointed out. Statistical adjustment for multiple comparisons were not conducted as there were not any confounding factors and no statistical difference in age between the groups. To minimize the need for multiple comparisons these patients were selected from a well defined Swedish InfCare cohort with >6000 patients where >99% of them were under treatment.

5. Only a very small subset of participant samples was included in the proteomics analysis. Is there any way for the authors to confirm the findings that apoptotic profiles may differ between PLWHEC and PLWHART, for example through flow cytometry staining for active caspase 3 expression or similar by CCR6⁺ and CCR6⁻ CD4 T cells?

Response: We thank the reviewer for pointing this out and ideally, we would have liked to perform this type of validation. We understand that the sample sized used in this cohort is low but in our case the experimental conditions was a limitation. In our initial flow cytometry analysis we saw that the % of CD4⁺CCR6⁺ cells varier from round 2% to 35% in the different individuals. Therefore, to obtain sufficient material for downstream analysis the starting material for the cell sorting was 10-20M cells per individual. This is a extremely high cell number to start off with for all individuals, and in our case we could only use what was at our disposal. On the other hand, our proteomic analysis detected decreased glycolysis and an enrichment of OXPHOS independent of CCR6 status and an enrichment of apoptosis and p53 signalling in CCR6⁺ cells in PLWH_{EC} compared to PLWH_{ART}. As seen in the figures below we tried to perform flow cytometry analysis to look at Activated Caspase 3 and p53 in the cohort. On request of the reviewer activated capsase 3 was investigated in the CD4⁺CCR6⁺ and CD4⁺CCR6⁻ cell populations (results below show results for CD4⁺CCR6⁺ and CD4⁺CCR6⁻ cells). Caspase 3 is an important marker of apoptotic cells that is can be activated by both intrinsic and extrinsic mediated pathways. As seen below, there was a

Response to Reviewers:

trend in increase in the median of activated caspase 3 positive cells in PLWH_{EC} compared to PLWH_{ART} in CD4⁺CCR6⁺ cells, although not significant. Additionally, we investigated the intracellular levels of p53 in the same cohort as it plays an important role in cell cycle progression, DNA stability, and apoptosis. As seen in the figure below, a slight decrease in the median % cells expressing p53 was detected in PLWH_{EC} compared to PLWH_{ART} in both CD4⁺CCR6⁺ and CD4⁺CCR6⁻ cells, although not significant. This data could indicate that PLWH_{EC} have higher apoptotic capacity through activated caspase 3 which is not mediated through p53 levels itself. Unfortunately, our sample availability was extremely limited (ART n=16 and EC n=7). Initially, we tried to recruit more samples but due to the COVID-19 pandemic recruitment of the patients was not successful. Therefore, we do not have any samples left from this cohort, that has been extensively used in other published articles from the group. As the sample size was so limited we do not consider this data to be sufficient to include in the study. However, we have now added the statement; “However, the sample size included in this study is relatively small and further studies validating these results are warranted” on line 435-436 in the discussion.

6. Figure 1b, Figure 2c, Figure 3a: The bubble plots are difficult to interpret, and it would be more helpful to be able to see the individual data points for each study participant, as well as error bars, similarly to how the data is shown in panel 1a.

Response: We thank the reviewer for this important comment. In each of section of the main figures (Figure 1c and d, and Figure 2d) we have now added the corresponding figure exhibiting % cells expressing a receptor (New figures can be viewed in question 9 by reviewer 1). These figures were previously available in the supplementary material but as suggested by the reviewer is better to include in the main figures for clarification (also as specified in question further down). However, we have kept the bubble plots as an overview just to show that the results for the other receptors, not included in the main figures, do not show any major differences. We believe the bubble plots can give an overview of what we have looked at so that the viewer quickly can see that there is no significant differences in the remaining receptor expressions.

7. Figures 1, 2, 3, and 5: Are the contour plots included in these figures from a PLWHEC, PLWHART, or HC? This should be specified in the figure legend.

Response: This is an important comment that we are thankful for the reviewer for pointing out and we have now tried to address the question. For Figure 1, and 2 the following statement has been included to clarify the origin of the contour plots; “Contour plot show a representing sample from PLWH_{EC}, PLWH_{ART}, and HC corresponding to the median % of cells within each group” in line 709-710 and 722-723, respectively. For Figure 3 the following statement has been included for clarification; “Contour plot show a representing sample from T1 and T2 corresponding to the median % of cells within each group” on line 739-740. In

Response to Reviewers:

figure 5, no contour plots were used. Herein, it is the gating strategy used for cell sorting that is exemplified by one of the HC samples. To clarify this in the figure legend we have now added the statement; "Gating strategy for isolation of CD4⁺CCR6⁺ and CD4⁺CCR6⁻ cell populations from total peripheral blood mononuclear cells, exhibited by an HC sample" On line 758-760.

8. Figure 5: The number of samples in each study participant group is missing from the figure legend.

Response: This is a mistake made from our side and we thank the reviewer for pointing it out. The number of samples used for each study group has now been stated in the figure legend for figure 5 as: "... cell populations isolated by FACS from PLWH_{EC} (*n*=4), PLWH_{ART} (*n*=6), and HC (*n*=3)" on line 757-758.

9. Several of the panels in the supplementary materials should be moved to the main text. In particular, the authors should consider moving Supplementary Figures 2a and b, Supplementary Figure 3b, and Supplementary Figure 5a, b and c to the main text.

Response: This is a good point that the reviewer has and we agree. To make the information more accessible we have now moved these figures from the supplementary into the main figures.

More specifically:

Supplementary Figure 2a and b have been moved to main Figure 1c and d, respectively.

Supplementary Figure 3b have been moved to Figure 2d.

Supplementary Figure 5a, b, and c have been moved to the main Figure 4b, c, and e, respectively.

The three updated figures can be viewed in order below.

Response to Reviewers:

Figure 1 Supplementary Figure 2a and b have been moved to main Figure 1c and d, respectively.

Response to Reviewers:

Figure 2 Supplementary Figure 3b have been moved to Figure 2d.

Response to Reviewers:

Figure 4 Supplementary Figure 5a, b, and c have been moved to the main Figure 4b, c, and e, respectively.

10. Lines 172-173: There appears to be a typo as the sentence should likely be “significant effect of CCR2 expression levels”, not CCR6.

Response: We thank the reviewer for noticing this error and have now changed it to saying the “significant effect of CCR6 expression levels” on line 317.

11. Lines 271-273: This sentence appears to be missing a reference.

Response: The reviewer is correct that a reference is missing in that line. We have now added in the reference Lee et al. (2017), CCR6/CCL20 chemokine axis in human immunodeficiency virus immunity and pathogenesis published in The Journal of general virology. Additionally, to clarify the message we have adjusted the sentence so it states; “CCL20 has previously been described to have an effect on the infection steps of HIV-1 and migration/function of immune cells” on line 414-416.

Reviewer #2 (Remarks to the Author):

The article by Akusjarvi et al investigates chemokine receptor expression on T cell and monocyte subsets from peripheral blood mononuclear cells in chronically infected HIV patients with viral loads below the detection level (<40 copies/mL). Patients were either elite controllers (EC; n = 14) or on anti-retroviral therapy (ART; n=54). A cohort of n =18, HIV negative individuals were also studied. The authors investigated the expression levels of CCR2, CCR3, CCR5, and CCR6 in T cells and monocytes by flow cytometry and performed latent reservoir analysis on bulk CD4 T cells. The data while interesting, are not presented in a logical cohesive manner and appear to be extremely fragmented. This in addition to the

Response to Reviewers:

lack of sufficient rigor significantly dampens enthusiasm for the study.

1. The flow of the Abstract is confusing as the first few lines imply an analysis focused on understanding the inflammatory consequences of HIV infection despite viral suppression. But the remainder of the Abstract then focusses on the phenotype of CD4 T cells in ECs.

Response: We would like to thank the reviewer for pointing out this important aspect of the manuscript and agree that the message was a bit misleading. We have now modified the abstract to give a broader description of our work. The modified abstract is as follows; "HIV-1 infection induces a chronic inflammatory environment not restored by suppressive antiretroviral therapy (ART). As of today, the effect of viral suppression and immune reconstitution in people living with HIV-1 (PLWH) has been well described but not completely understood. Herein, we show how PLWH who naturally control the virus (PLWH_{EC}) have a reduced proportion of CD4⁺CCR6⁺ and CD8⁺CCR6⁺ cells compared to PLWH on suppressive ART (PLWH_{ART}) and HIV-1 negative controls (HC). Expression of CCR2 was reduced on both CD4⁺, CD8⁺, CM, and IM monocytes in PLWH_{EC} compared to PLWH_{ART} and HC. Longer suppressive therapy, measured in the same patients, decreased number of cells expressing CCR2 on all monocytic cell populations while expression on CD8⁺ T cells increased. Furthermore, the CD4⁺CCR6⁺/CCR6⁻ cells exhibited a unique proteomic profile with a modulated energy metabolism in PLWH_{EC} compared to PLWH_{ART} independent of CCR6 status. The CD4⁺CCR6⁺ cells also showed an enrichment in proteins involved in apoptosis and p53 signalling in PLWH_{EC} compared to PLWH_{ART}, indicative of increased sensitivity towards cell death mechanisms. Collectively, this data shows how PLWH_{EC} have a unique chemokine receptor profile that may aid in facilitating natural control of HIV-1 infection" on line 31-44.

2. Figure 1. The authors should clarify the parent gate that CD4 and CD8 T cells derived from - % of what subset? The total appears to be greater than 100.

CD4 T cells should be sub-setted by naive and memory subsets

The authors show MFI differences in CCR2 and CCR6 – since MFI is highly dependent on laser intensity and essentially the intensity can vary from day-to-day, what measures were taken to ensure consistency? Was an internal control used?

Enumerating frequency of CCR2 and CCR6+ cells would be informative.

Response: We would like to thank the reviewer for pointing this out these concerns to us and we have now tried our best to address them. In Figure 1a, we have clarified that the CD4 and CD8 T cells are derived from CD3⁺ cell (CD3⁺CD4⁺ T cells (%) or CD3⁺CD8⁺ T cells (%)). To make it more clear we have also included the statement " derived from CD3⁺ cells" in the figure legend for Figure 1a on line 703. Regarding the frequency of these cells, we do see that there is a large patient to patient variation of % of cells and understand that at a glance it can look like it exceeds 100% but collectively when combining the cells per patient together the CD4 and CD8 T cells are not more than 100%.

Concerning the sub-setting of CD4 T cells, this is a question we would have liked to answer but unfortunately due to low sample availability this is not possible. We initially aimed to recruit more patients for sample collection but due to the COVID-19 pandemic this has not been possible.

Regarding the MFI, no specific internal control was used. But when analysing the data for the T1 and T2 samples, samples from the same individual, except for one case, was run in the same batch to minimize any experimental variations. Regarding the other differences the HC samples were divided in the different runs. This was performed so that any differences between the runs would then be detected within the spread of the HC itself. Following, as the reviewer stated that enumerating the frequency of CCR2⁺ and CCR6⁺ cells would be more informative we have now included this information into the main figure (Figure 1c and d, and Figure 2d), as seen in the figures above for question 9 from Reviewer 1. The data for enumerating the frequencies of CCR2 and CCR6 positive cell was originally in the

Response to Reviewers:

supplementary figure but as pointed out by both reviewers are more suitable in the main figures.

3. Figure 2. Similar to expression profile in T cells, was the frequency of CCR2- expressing monocytes altered in HIV infection? Quantifying the ligands for CCR2 and CCR6 would have been informative.

Response: The reviewer has a good point that plasma levels of the ligands of CCR2 and CCR6 is a informative to this study. The data for quantification of plasma levels of the ligands for CCR6 (CCL20) and CCR2 (CCL2) is available in Supplementary Figure 2c and d (Corresponding to Supplementary figure 2e and f in the initial submitted draft).

As for CCR2 expression on the monocytic cell populations we showed the % cells expressing CCR2 (CCR2⁺) in the manuscript. To show the results more clearly we have now moved the figure showing % of monocytes expressing CCR2 from Supplementary Figure 3 (supplementary figure 3b in the initially submitted manuscript) to Figure 2d (as seen in figure above in question 9 for Reviewer 1). To clarify for the reviewer we have here now included the results for CCR2⁻ expressing monocytes on CM, IM, and NCM (see figure below). As seen, there is no significant difference in CCR2⁻ cells on CM while CCR2⁻ cells are increased in both PLWH_{EC} and PLWH_{ART} compared to HC on IM. Furthermore, on NCM there is a significant increase in CCR2⁻ cells in PLWH_{EC} in relation to PLWH_{ART}. In conclusion, this data shows that the frequency of CCR2⁻ cells vary on some subpopulations during controlled HIV-1 infection, either treatment or spontaneously induced.

4. Figure 3 shows longitudinal increase in MFI and frequencies of CCR2 and CCR6 on CD8 T cells after 3 years. What was the viral load data at T1 and T2?

Response: We thank the reviewer for this question. As stated in Supplementary Table 2, there is no difference in viral load between T1 and T2. In both the time the VLs were below detection level.

5. Measurement of the HIV reservoir across EC and ART suppressed patients could have been performed on CCR6+ versus CCR6- CD4 T cells.

Response: We agree with the reviewer and would like to thank for pointing this out. HIV reservoir measurement was performed in the separated CCR6⁺ and CCR6⁻ CD4⁺ T cells. This data is available in supplementary figure 6a. Unfortunately, as the sample size was small for the cell sorting experiment no significance was detected between these two cell populations. Initially, we aimed to recruit more patients to increase the sample size but due to the COVID-19 pandemic this is not possible. Therefore, we are limited to the small sample size available. In our cohort, the % of CD4⁺ T cells expressing CCR6 varied between round 2% to 35% in the individuals, which shows a large donor variation. This comes with its challenges as cell sorting then required us to start with a large number of cells to be able to isolate sufficient cells for downstream analysis. In our case, we started from 10-20M cells and could not obtain this from all PLWH_{EC} donors.

REVIEWERS' COMMENTS:

Reviewer #1 (Remarks to the Author):

In this revision, the authors have addressed the majority of my comments and I believe that the manuscript is improved. I would like to thank the authors for their careful and detailed responses. Of note, I would like to clarify my 4th question on adjustments for multiple comparisons. Here, I was curious if measures were taken to avoid Type 1 errors in the statistical analyses. For example, in the methods, under "Bioinformatics analysis" line 223, it is stated that the Benjamini-Hochberg method was used to control the false discovery rate in the gene-set analysis. However, I was curious which, if any, statistical methods were used for the remaining statistical analyses, particularly for the flow cytometry data, to avoid Type 1 errors?

Reviewer #2 (Remarks to the Author):

The authors have satisfactorily addressed my concerns.

REVIEWERS' COMMENTS:

Reviewer #1 (Remarks to the Author):

In this revision, the authors have addressed the majority of my comments and I believe that the manuscript is improved. I would like to thank the authors for their careful and detailed responses. Of note, I would like to clarify my 4th question on adjustments for multiple comparisons. Here, I was curious if measures were taken to avoid Type 1 errors in the statistical analyses. For example, in the methods, under "Bioinformatics analysis" line 223, it is stated that the Benjamini-Hochberg method was used to control the false discovery rate in the gene-set analysis. However, I was curious which, if any, statistical methods were used for the remaining statistical analyses, particularly for the flow cytometry data, to avoid Type 1 errors?

Reply: The reviewer has brought up an important question. Regarding the statistical analysis for the flow cytometry data, no specific method was employed to avoid Type 1 errors. We considered the expression of each chemokine receptor to be independent of one another. Therefore, we did not feel the need to apply a post hoc correction for multiple comparisons such as the Bonferroni test. We treated all the chemokine receptor expression as independent of one another in our analysis. In the correlation analysis, we also evaluated if these receptors potentially were co-expressed with one another but as discovered the cell expression did not always correlate. Therefore, we do not consider them to be co-dependent and this type of statistical method not necessary.

Reviewer #2 (Remarks to the Author):

The authors have satisfactorily addressed my concerns.

We would like to take the opportunity to thank the reviewer for their valuable comments to improve our manuscript.